# Effective Survey Methods for the Elusive Data Deficient Black Flying Squirrel (*Aeromys tephromelas*) in Sabah, Malaysia Facilitate First Vocalisation Record

**DOI:** 10.3390/ani14223323

**Published:** 2024-11-19

**Authors:** Sapphire Hampshire, Priscillia Miard

**Affiliations:** 1Night Spotting Project, Kota Kinabalu 88400, Sabah, Malaysia; nightspottingp@gmail.com; 2Department of Conservation Biology, Georg-August-University Goettingen, Wilhelmsplatz 1, 37073 Goettingen, Germany; 3Institute for Tropical Biology and Conservation, Universiti Malaysia Sabah, Jalan UMS, Kota Kinabalu 88400, Sabah, Malaysia

**Keywords:** giant flying squirrels, Borneo, Pteromyini, bioacoustics

## Abstract

Flying squirrels are nocturnal gliding mammals native to forest habitat and range broadly in size from 24 g up to 1.5 kg. There are 52 species worldwide, 95% of which are native to Asia, yet most research focuses on the three species found in North America. With minimal data on Asiatic flying squirrels, obtaining ecological data is key for improving conservation efforts. In February and March of 2023, a nocturnal survey was conducted at the Rainforest Discovery Centre (RDC), at the edge of primary forest in Sepilok, Sabah. To improve detection chance, the study included multiple specialised monitoring equipment including red-light headtorches, a thermal camera and an audible-ultrasonic microphone. Three giant flying squirrel species were observed, most notably the black flying squirrel, which is so understudied that its extinction risk has not been assessed. This study also provided the first documented vocalisation event for this species, with 106 calls within a frequency range of 0.75–2.69 kHz and an average duration of 1.4 s. With flying squirrels reliant on trees, deforestation across their distribution poses a major threat. Therefore, this study both highlights the urgency for assessing the black flying squirrel’s extinction risk and understanding their role in the ecosystem.

## 1. Introduction

Flying squirrels, tribe Pteromyini (Brandt, 1855), are the nocturnal relatives of their non-gliding counterparts within the order Sciuridae [1]. Their fore- and hind-limbs are attached to a large membrane (patagium) used for gliding from tree to tree, with the use of their tails for steering [2]. Their size ranges greatly, from ~24 g (*Petaurillus kinlochii*) to ~1539 g (*Petaurista petaurista*) [2]. No matter their size, squirrels utilise tree cavities for sleeping, canopies for gliding and have predominantly folivorous diets [1].

There are currently 52 species of flying squirrel recognised by the IUCN Red List, with the greatest diversity (95%) found across Asia [3]. Despite this, when a literature review on publications was conducted in 2006, it showed a major skew in the literature with >265 publications on the North American flying squirrels (*Glaucomys* spp.) and >60 for the Siberian flying squirrel (*Pteromys volans*) [1]. Contrastingly, that study found that 80% of Asiatic species have <10 publications, with at least 10 species having no focused literature, which remains the case for many flying squirrel species [1]. Within Asia, Malaysia has the highest number of species (17), 11 of which reside in the Malaysian state of Sabah (Table 1). Sabah (73,371 km^2^) is situated on the island of Borneo bordering both Sarawak (Malaysia) to the west and Kalimantan (Indonesia) to the south [4,5].

Of the flying squirrel species in Sabah, the IUCN Red List has categorised 27% as either Endangered or Vulnerable and 27% as Data Deficient [6,7,8,9,10,11,12,13,14,15,16]. The remaining 45% often fly under the radar by being classified as Least Concern, when there is minimal population data available. This makes many flying squirrels susceptible to local extinctions as the early warning signs are easily missed [17,18]. Fortunately, of the 11 species, nine are protected under Schedule II (protected and limited hunting with a license) of the Wildlife Conservation Enactment 1997 [19]. However, one of the unprotected species is the Jentink’s flying squirrel, currently considered Data Deficient. Currently, there are no focused publications on flying squirrels in Sabah, leaving a large gap in our understanding of their ecology [1]. The four giant flying squirrel species have been noted as present from rapid assessment surveys and species checklists across Sabah, yet occurrence data is still limited by the lack of areas assessed (<10% of forest reserves) and predominantly from diurnal surveying (Figure 1). In four areas there is the reported presence of multiple giant flying squirrel species, yet there is no published information on their sympatric relationships [20,21,22,23,24,25]. Acknowledging co-existing species with similar ecology is important to understand resource partitioning. In addition to impacting the carrying capacity [26], niche differentiation can develop, altering species ecology such as feeding strategies [27].

Out of the giant (>30 cm body length) flying squirrel species, the enigmatic black flying squirrel (*Aeromys tephromelas*) remains elusive, classified as Data Deficient by the IUCN Red List [6,32]. Identifiable by the deep black colouration of its hind pelage and lighter underside, it has a combined head and body length of 35–42 cm [2,32]. Whilst similar in size to its relative, the Thomas’ flying squirrel (*A. thomasi*), *A. tephromelas* weighs slightly less at around 1.2 kg, but has a much longer tail length [32]. Their current distribution around Sabah is poorly known, but their presence has been documented in Kinabalu Park [23,32], the Danum Valley Conservation Area [34], Maliau Basin Conservation Area and the Mt. Louisa Forest Reserve [20]. Additionally, in 2017, a National Geographic article published camera trap footage of *A. tephromelas* at the Rainforest Discovery Centre in Sepilok [24]. With such paucity in population data, even less is known of their behavioural ecology, which is constrained by the difficulty of nocturnal monitoring.

Despite making up 79% of Sabah’s mammals, nocturnal and crepuscular species such as flying squirrels are highly underrepresented in the literature [35]. Folivorous and frugivorous species often play a fundamental role in the ecosystem, acting as seed dispersers [36]. Surveying nocturnal mammals is notoriously challenging with dense vegetation, reduced visibility and hard-to-navigate terrain impeding movement and reducing field safety [37,38,39]. To overcome these limitations, surveys are often high in effort (many sites, multiple repetitions), yet detection rates of many species remain low [40,41]. For example, 84 field days only yielded 12 observations of the red giant flying squirrel (*Petaurista petaurista*) on Langkawi Island (Malaysia) [41]. In Sabah, there are 333 protected wildlife areas and forest reserves, but <10% have published data on mammal assemblages [20,25,28,30,33,42,43]. Consequently, population and ecological data, key to ensuring effective management in protected areas, is scarce for many species [44].

With over-population, resource demand and the climate crisis creating a time limit for reducing extinction risk, improving detection rates for nocturnal species is crucial. Therefore, researchers are constantly striving to improve nocturnal surveying methodologies, and the advances of technology have presented thermal imaging, camera-trapping and acoustic monitoring as non-invasive, cost-effective solutions to the nocturnal surveying dilemma [45,46,47,48,49]. Additionally, the effectiveness of white spotlighting has been questioned as it is known to dazzle wildlife, trigger flight responses, alter their observed behaviour, and/or cause damage to their eyes [37,50]. Red spotlights have been used as an alternative to reduce disturbance [37], as most nocturnal mammals are unable to perceive wavelengths higher than approximately 580 nm, yet red light surpasses this at 625–740 nm [51,52,53].

Mammals use acoustic communication for a multitude of reasons including orientation, defence, and social cohesion [54]. In recent years, the use of bioacoustics has surged, becoming a prominent tool for monitoring species occurrence, allowing for increased data collection for cryptic species and in areas where observer visibility is reduced, such as the rainforest [55]. As rainforests are naturally auditory environments with multiple species concurrently vocalising, bioacoustic monitoring can still be challenging as observers must focus on selective signals, easily masked by the rainforest chorus [56,57]. However, advances in specialised recording devices increase the breadth of bioacoustic possibilities. For example, ultrasonic calls discovered in the three American flying squirrel species (*Glaucomys sabrinus*, *G. volans* and *G. orengonesis*) have allowed for effective bioacoustic monitoring to combat the difficulty of following fast moving, high-dwelling flying squirrels [58,59]. Currently, there is limited research on Asiatic flying squirrel repertories, yet Newar and Bowman (2020) suggest the potential for ultrasonic calls as a niche communication channel, as nocturnality is directly associated with higher frequency vocalisations [60]. Surveying with equipment that records both audible sound (20–20 kHz) and ultrasound (>20 kHz) could be a useful nocturnal monitoring tool, both directly for ground surveying techniques, as well as for passive acoustic monitoring [61].

Between February and March 2023, a nocturnal mammal study at the Rainforest Discovery Centre in Sepilok was conducted to gain a baseline inventory of the species at the centre, as well as to evaluate the use of red spotlighting, thermal imaging and bioacoustic recording for nocturnal monitoring, as recommended by Miard et al. 2024 [41].

This survey revealed the presence of multiple flying squirrel species and therefore, this paper addresses the following questions: (i) Which giant flying squirrels were present in the field site? This is pertinent for inventorying species, as well as for unveiling any sympatric relationships. (ii) Which monitoring method was most efficient for detection? Due to their bright eyeshine and fast-moving nature, it is expected that flying squirrels are best detected by red spotlighting. (iii) Which behaviours are present when first sighted? Behaviour at first sight can indicate disturbance from the survey and tolerance levels between different species may vary. (iv) How do the different species of flying squirrels utilise their habitats at the field site? It is expected that different species may favour different canopy layers indicating niche partitioning to reduce competition. (v) If observed feeding, what were the dietary preferences of these nocturnal species? As flying squirrels are omnivores, it is expected that a diversity of items are consumed, but with a predominance of plant parts.

## 2. Materials and Methods

### 2.1. Study Site

This study took place at the Rainforest Discovery Centre (RDC) within the small village of Sepilok in Sabah, Malaysian Borneo (Figure 2). Opened to the public in 2007, the RDC is one of the most notable ecotourism centres in Sabah, considered a model facility for environmental education, conservation of tropical rainforests and ecotourism [62,63]. The RDC is located in the 106 ha Taman Botanikal Sepilok Forest Reserve area, which constitutes the edge of the Kabili-Sepilok Forest Reserve (4326 ha) [63]. The vegetation is lowland (~0–100 masl) dipterocarp primary forest (Class VI Virgin Jungle) and is accessible via 8 ground trails through the forest as well as the 25 m high canopy walkway [62,64].

### 2.2. Survey Parameters

Surveying was conducted between February and March 2023 during the monsoon season [65]. The sampling period ran between 19:30 and 00:00, to align with a previous nocturnal mammal study on Langkawi Island and Penang Island [41]. The sampling start time was chosen to avoid night-walks (18:00–19:00) run by the RDC where multiple white spotlights are used with large groups of tourists on many of the trails. There were two survey techniques: (1) point counts and (2) opportunistic observations. Between points observers used a pace of one km/h [66] so that opportunistic sightings could be included to increase detection of wildlife [46]. To minimise disturbance to the wildlife, systematic along-transect point counts were chosen [67]. Therefore, the nine pre-existing trails were split into 10 transects (Figure 3), with all except two satisfying the minimum length of 500 m for a transect, as recommended by Miard et al. (2024) [41]. Points were spaced at 100 m distance, totalling 57 across the 10 transects. Point observations consisted of observers standing and scanning 360° around the point for 10 min, with a maximum number of three observers chosen to reduce human-induced flight responses of the wildlife [68]. There were three repeats per point and repetitions were never consecutive to minimise disturbance [66,69].

### 2.3. Data Collection

There were three main pieces of equipment used for the surveys: (1) Red spotlights, (2) hand-held thermal cameras, and (3) ultrasound/audible sound microphones (Table 2). Traditional white spotlighting can dazzle wildlife, damage their eyes and trigger flight responses [37,50]. Red light is considered the best alternative as it has long wavelengths that most nocturnal mammals cannot perceive [70]. Thermal imaging was included to reduce the reliance on catching eye-shine from an animal [71]. As many nocturnal species have developed ultrasonic calls as a niche communication channel [72], an ultrasonic microphone with a directional horn was used, which recorded at a sampling frequency of 384 kHz, able to detect both audible and ultrasonic sound up to 150 kHz [73,74].

Sighting, GPS and environmental data (rain, cloud cover, temperature, humidity) were recorded at the points and opportunistically using a custom Google© AppSheet phone application (https://about.appsheet.com/home/). Acoustic data were collected through the microphone attached to a smartphone using the Bat Recorder application (© Bill Kraus). For the points, the microphone continuously recorded but was also used during any opportunistic sightings. Any acoustic recordings were accompanied by a video recording, if possible, to ensure correct identification of the vocalisation origin.

Behaviour at first sight was included to assess disturbance from the observers. Flying squirrel behaviours were defined using ethograms and behavioural literature (Table 3) [75,76].

To assess stratification use of the canopy, observations were categorised into the following vegetation levels [77,78,79]:Emergent: >30 mUpper Canopy: 23–29 mLower Canopy: 15–22 mUnderstory: 6–14 mUndergrowth: 0–5 m

When feeding observations were made, trees were marked and leaves/branches were collected for the Rainforest Discovery Centre arboretum to assess and provide species-level identification.

### 2.4. Data Analysis

Observation frequency for both point and opportunistic sightings were reported for all three giant flying squirrel species. Naïve occupancy estimates were calculated as the proportion of transects where the species was detected [80]. First detection by eye-shine (red spotlight), thermal imaging and ultrasonic were included with the observation data.

The results for behaviour, vegetation use and feeding of all three giant flying squirrel species were reported. Behaviour at first sight was plotted as a cluster bar chart with the proportion (%) of behaviour for each flying squirrel species. For vegetation data, only descriptive statistics (bar charts) were used as the sample size was too low for effective comparisons through inferential statistical tests. The differences between species feeding trees and plant parts were qualitatively analysed and discussed.

The black flying squirrel audible call was analysed in RAVEN Pro 1.6.5 to produce a spectrogram, where the DFT size and windowing function are reported. The call parameters measured were: Start frequency (kHz) and End frequency (kHz), with the call duration(s) calculated from these. Highest and lowest frequency (kHz) were measured as well as peak frequency (kHz) and the bandwidth at 90% (kHz).

## 3. Results

This survey revealed the presence of three of the giant flying squirrel species: the red giant flying squirrel (*Petaurista petaurista*), the Thomas’ flying squirrel (*Aeromys thomasi*) and most notably the Data Deficient black flying squirrel (*A. tephromelas*). With a paucity of data, this paper provides the first descriptive report of black flying squirrel observations, including the first documented vocalisation data for the species.

### 3.1. Observations

Of the 98 observations of mammals at the RDC, there were nine observations of black flying squirrels (*Aeromys tephromelas*) (Figure 4), two of Thomas’ flying squirrels and 19 of the red giant flying squirrels.

Two other giant flying squirrel species were sighted: the red giant flying squirrel (*Petaurista petaurista*) (*n* = 19) and Thomas’ flying squirrel (*Aeromys thomasi*) (*n* = 2) (Table 4). There were 12 observations of small flying squirrels, but none were identified to species-level; thus, are not reported here.

The naïve transect occupancy for *Aeromys tephromelas* was 40%, with 55% of sightings along the ridge trail, whilst *A. thomasi* was found on the Canopy Walkway on both occasions. For *P. petaurista*, the naïve occupancy was 70%, with the most frequent locations being the Canopy Walkway (32%) and Mousedeer Crossing (26%).

Of the six point observations for *Aeromys tephromelas*, all but one were first detected by eyeshine using red spotlight, the same result as for *Petaurista petaurista*. For *Aeromys thomasi*, the first detection was by eyeshine whilst the other was by thermal imaging. Opportunistic observations were 100% eyeshine as thermal was not consistently used when walking between points.

The five behaviours identified were feeding, foraging, grooming, moving, and observing, with three observations of *Petaurista petaurista* marked as unknown. Behaviour at first sight (%) was plotted for *Aeromys tephromelas* and *P. petaurista* (Figure 5). Movement was the top proportion (55%) for the black flying squirrel (*Aeromys tephromelas*), whereas observing was the highest for the red giant flying squirrel. *A. thomasi* was the only squirrel observed grooming when first sighted, with their other first sighting being a feeding event.

The proportion of observations within the different vegetation strata were plotted for the three species (Figure 6). The black flying squirrel was predominantly found in two of the lower canopy levels (66% understory and 22% lower canopy), whilst the two observations of Thomas’ flying squirrel were only in the upper canopy. The red giant flying squirrel was observed almost equally (21–26%) across all strata except the undergrowth (5%).

### 3.2. Feeding

During the survey, all three giant flying squirrels were observed feeding on five different tree species (Table 5). *Aeromys tephromelas* and *A. thomasi* both fed on plants in the *Meliaceae* family, but differing species. Additionally, the Thomas’ flying squirrel was observed eating from a Dipterocarp tree during the second observation. The red giant flying squirrels were observed eating leaves from *Palaquium dasphyllum*, but the food item could not be determined in the *Neolamarckia cadamba* tree.

### 3.3. Bioacoustics

On the 7 March 2024 at 21:11 an ad hoc recording of a black flying squirrel audible call was achieved on the Ridge trail (Figure 3). The individual was video recorded to validate the origin of the call [81]. There were two individuals, one at 20 m high in a tree, another at a height of 9 m in a different tree approximately 5 m from the prior. The first individual was continuously vocalising for approximately 15 min within earshot of the observers. The vocalisation event was recorded for 12 min, but only 3 min and 20 s could be analysed due to a corruption in the sound file. The second individual glided out of sight 8 min into the recording, but the vocalising continued for a further 4 min.

During the 3.33 min, 106 calls were made by the individual, with 60 clear enough for analysis. The call structure was isolated and categorised as a chirp (Appendix A). Each call consisted of two components, with the first being notably longer and stronger in amplitude (Figure 7).

Call parameters were calculated (Table 6) with calls ranging between 0.75 kHz and 2.69 kHz, with an average frequency variation of 1.19 kHz per call. Call duration ranged between 0.08 and 0.18 s, averaging 0.14 s per call. Peak frequency ranged between 1.89 kHz and 2.58 kHz with an average of 1.97 kHz. During the recording there were eight bouts of calls (defined as gap between calls of >4 s. Repetition rate (time between calls) ranged between 0.4–2.5 s, averaging 1.46 s across all bouts.

## 4. Discussion

Understanding the ecology of a species is integral to assessing their role in the ecosystem, subsequently shaping effective conservation strategies [82]. This study presents the first ecological data on wild black flying squirrels this side of the millennium. Key results are the first documentation of *Aeromys tephromelas* vocalising and the confirmed co-existence with two other giant flying squirrels (*Petaurista petaurista* and *Aeromys thomasi*). Given these findings, the Rainforest Discovery Centre (Sepilok, Sabah) emerges as a critical site for further research on flying squirrels, providing essential opportunities to deepen our understanding of their ecology and inform conservation efforts.

### 4.1. Observations

Prior to this study, the only published presence of black flying squirrels within the Kabili-Sepilok Forest reserve was from camera trap footage at the RDC in a National Geographic article [24]. Their presence has been long known to the staff guiding night-walks, an ecotourism initiative run by the RDC. However, when asked, they commented that they rarely see them on night-walks [83], which is interesting as 90% of observations were on trails typically used during night-walks. Understanding why detection was higher in this study could have positive implications for improving monitoring methods for future surveys.

One factor could be the timing of the survey. At the RDC, night-walks start at 18:00 to enhance tourist’s chances of spotting wildlife as colugos (*Galeopterus variegatus*) and red giant flying squirrels (*Petaurista petaurista*) become active before sunset [83]. Understanding a species’ activity budget can help explain behavioural adaptations to their environment [84]. For giant flying squirrels, the only activity budgets published focus on the Indian giant flying squirrel (*P. philippensis*), which has two nightly peaks between 19:00–20:00 and 03:00–04:00 [84,85]. For the black flying squirrel, the nine observations varied between 20:00 and 23:00, which could indicate their activity peaks differ either naturally, or to adapt to disturbance from the night-walks. However, with only nine observations and the survey time starting after the night-walks, there is not enough data to draw any conclusion.

Response to moonlight could also impact flying squirrel activity as owl monkeys (*Aotus* spp.) have a peak of activity during twilight [86], whilst slow lorises (*Nycticebus* spp.) are typically lunarphobic [87]. The only mention about flying squirrel activity and the influence of moonlight in Asia is for *Eoglaucomys fimbriatus* (Kashmir), where they hypothesised that moonlight did not impact their activity [88]. Observations of *Aeromys tephromelas* were found in varying stages of moonlight, with 66% during the waning gibbous (80–97% disc illumination). However, two observations were at lower illumination (~30%) (Appendix A). Sample sizes were too low for statistical analysis on the effect of moonlight but should be considered in future studies as it could improve the efficacy of monitoring cryptic species.

For the first detection of all species in the study, including the black flying squirrel, red spotlighting was more effective than thermal imaging, which was also found by Miard and her team (2020) in Peninsular Malaysia and Brunei Darussalam [89]. However, thermal imaging should not be discounted as an aid in surveys as it was particularly useful for observing behaviour when individuals were obscured by foliage or at far distances [90]. Most observations of *Aeromys tephromelas* were in the understorey level of the canopy where they occupied the top third of those trees. Although their height in the tree was only taken at first sighting, they rarely moved far, with a tendency to forage on the same tree for the duration of the observation or move to the top third of another tree. There were ad hoc sightings of *A. tephromelas* in higher canopy levels, which could suggest either/both an observer bias, as it is simpler and easier to see the lower canopy levels, or a location bias [91]. For example, the observers were using wide preexisting trails, which often have decreased canopy cover and allow for increased leaf production, potentially providing more food sources and higher observer visibility [92]. These factors should be accounted for in future studies to avoid potential pseudoreplication and a misrepresentation of occupancy [93].

Finally, one major difference from the night-walks was the use of red spotlights instead of white spotlights. As this study did not include comparative data for white spotlights, it is inconclusive as to whether it impacted detection. In both Peninsular Malaysia and Brunei Darussalam, detection of nocturnal mammals increased by 46% using red spotlights compared to white spotlights [89]. Additionally, white spotlights have been found to negatively affect flying squirrels, for example, causing them to fall mid-glide [94]. Only one anecdotal observation occurred within this survey where during an identification check of *A. tephromelas*, when the light was switched from red to white, the individual immediately glided out of sight, which could have been a flight response. Gliding is theorised as an anti-predation technique allowing a fast get away [95]. Although the gliding distance of the black flying squirrel has not been studied, their cousins, the red giant flying squirrel (*Petaurista petaurista*) and the Indian giant flying squirrel (*P. philippensis*), frequently glide up to 20 metres, with the capability of 50–90 m [84,96]. Other than under white light, no fleeing behaviour was recorded during this study, but with the low observation count this does not provide enough evidence for reduced disturbance. However, reducing flight responses could be vital to improving the detection rate of these elusive gliders. With more knowledge emerging on red spotlights as a tool for improved detection and welfare [41,89], incorporating a comparative study of spotlights for black flying squirrels is highly recommended. Not only would this improve nocturnal surveying methodologies for researchers but could also contribute to nocturnal ecotourism efforts. For example, from June 2024, all night-walks at the RDC now only use red spotlights as a result of the surveying and recent research on their advantages over white spotlights.

### 4.2. Feeding Ecology

Most flying squirrels are considered generalist folivores due to their enlarged ceca and gut biota, synchronous with other folivores [97]. The red giant flying squirrel and Thomas’ flying squirrel both have flattened teeth, specialised for grinding [98,99]. However, the black flying squirrel has more cuspidate teeth, more similar to the smaller flying squirrel species, yet the reasons are unknown and there is minimal data to illuminate whether this impacts their diet [99]. There is a scarcity of data on black flying squirrel diets, with the most recent reference being Thorington and Koprowski (2012) stating their diet is seeds, fruits and plant material [100].

During the survey, feeding was observed from two trees in the Mahogany (*Meliaceae*) family: *Heynea trijuga* and *Azadirachta excelsa.* For the former, the individual was observed eating the buds of the fruit and/or flowers, whilst for the latter, the footage was too far to identify which plant part was ingested. On both trees the individuals’ cycled between pulling the branch and eating directly from it, as well as pulling the bud/unidentified plant part from the branch, then grasping the food with one paw or two paws to eat from. These behaviours were also present when *Petaurista petaurista* and *Aeromys thomasi* were observed feeding, unsurprising as food handling is very common in Sciuridae [101]. However, they fed at different trees: (1) *Petaurista petaurista*—Rubianceae: *Neolamarckia cadamba* and *Saptoaceae: Palaquium dasyphyllum*, (2) *Aeromys thomasi—Dipterocarpaceae: Parashorea tomentella*.

Feeding preference has been displayed by some giant flying squirrels already, with the Japanese flying squirrel (*Petaurista leucogenys*) switching between leaves and seeds to coincide with their reproductive cycle [102]. The Indian giant flying squirrel (*P. philippensis*) in the Western Ghats prefers fruit from Ficus species [103], whereas in Taiwan, young leaves and buds are preferred [85]. In Taiwan, *P. petaurista* predominately utilised buds and bark [104], yet in this study only leaves were observed being consumed. However, this difference could easily be because of the low number of feeding occasions observed. Understanding preferences can provide insight into their seed dispersing role in the forest [105]. A study at the Tabin wildlife reserve (Sabah) showed that civets showed non-random seed dispersal in comparison to macaques, showing they play a fundamental role [106]. Therefore, knowing the food preferences of *Aeromys tephromelas* and its relatives could help with both understanding resource partitioning and to help with conservation efforts (e.g., for reforestation and protecting areas).

### 4.3. Bioacoustic Communication

Here, the first call structure from a black flying squirrel is described and categorized as a chirp. The frequency range of this call (0.75–2.6 kHz) and peak frequency (av. 1.97 kHz) are particularly interesting as they lie just under the lowest frequency contributors to the rainforest chorus (2.5–3 kHz) at the RDC. This could be an evolutionary adaptation to ensure calls can be heard, as lower frequencies often have better sound transmission over short distances in the rainforest [107]. Lower frequency calls are also found in the Japanese giant flying squirrel (*Petaurista leucogenys*), between 1 and 15 kHz [108], and the White-faced flying squirrel (*Petaurista alborufus lena*), with two call types peaking around 3 kHz [109]. Additionally other rodents produce lower frequency calls such as Unita ground squirrels (*Citellus armatus*), which have agonistic calls ranging between 2–9 kHz [110]. It is important to note that more recordings could help isolate the call more accurately. For example, the top frequency of some of the calls could have been higher than analysed due to the overlap with the rainforest chorus of frogs and insects [111].

Previous vocalisation studies of flying squirrels are focused on *Glaucomys* spp. in North America, where their repertoires have been extensively documented [59]. Currently, 63% of flying squirrels have no recorded repertoire and species with calls are mostly descriptions based on either their onomatopoeic sound e.g., the “Scree” call from Hodgson’s giant flying squirrel (*Petaurista magnificus*) [112]. Some are even described by their pitch, e.g., Koli and Bhatnagar (2016) described *P. petaurista* as emitting low-pitched whistles and *P. alborufus* as emitting high-pitched whistles [84]. Although calling activity for the Indian giant flying squirrel (*P. philippensis*) was studied, no audio was analysed [84]. Therefore, Terada et al. (2021) is one of the first to analyse call structure for an Asiatic flying squirrel (*P. leucogenys*) [108]. With many flying squirrel habitats difficult to traverse, bioacoustic monitoring could be a useful way to map their occurrence, with studies on the American species (*Glaucomys* spp.) already at play [113]. Additionally, play-back experiments have been used to determine call functions for the Japanese flying squirrel [108].

The vocalisation event was the only time two individuals were seen at the same point. It is unclear which sex the animals were, as although females are slightly larger [114], this was not discernible during the observation. There was no audible response from the other flying squirrel, nor visible behavioural response before it glided out of sight approximately 4 min before the vocalising stopped. A lack of behavioural response aligns with 42% of flying squirrel species with known calls having no associated behaviour during vocalisations [59]. However, for many species, there is limited data on behavioural observations alongside vocalisations and thus it does not mean that calls do not have an associated behaviour. Therefore, ensuring behavioural observations are recorded in conjunction with vocalisations is an important step for better understanding of flying squirrel communication.

The call function cannot be confirmed, although the context implies the call could have been social, albeit with no context clues as to whether it was affiliative (e.g., courting) or antagonistic (e.g., territory defence). It appears unlikely that it was an alarm as it started before the squirrel reached the vicinity and ended whilst the vocalising squirrel was still being observed with no visible distress behaviours displayed. 

Despite the red giant flying squirrel having described calls [104], none were actively heard during this survey. This is likely explained by most observations being in the upper canopy and too far away to be heard over the rainforest chorus. Additionally, no vocalisations were heard from the Thomas’ flying squirrel, which is likely due to only seeing them twice. Although this is the first documented report of a black flying squirrel vocalising, these audible calls are already known and are identifiable by experienced guides at the centre, highlighting the importance of utilising local knowledge in research [83]. Much remains unknown about flying squirrel vocalisations, and the discovery of this call is a stepping stone to a better understanding for *Aeromys tephromelas.* Not only could it prove crucial for understanding social behaviour, but it could be used as a monitoring tool for black flying squirrel occurrence, which is urgently required.

### 4.4. Sympatry

Understanding a species’ ecology is a gateway to understanding their role in the ecosystem [82]. The Rainforest Discovery Centre has three of the four giant flying squirrel species, with there also being scope for small flying squirrel research. Their co-existence has been established in other surveys with all of the giant flying squirrels found in Kinabalu Park [21,23,115] and *Aeromys tephromelas*, *A. thomasi* and *Petaurista petaurista* have been found in both Mt. Louisa and Maliau Basin [20]. During this survey, individuals of both *Aeromys thomasi* and *Petaurista petaurista* were observed utilising the same dipterocarp (*Parashorea tomentella*). Habitat partitioning often occurs when species co-exist, such as between grey and red squirrels [116] and members of the *Glaucomys* genus [27]. In the Himalayas, *P. petaurista* and *Eoglaucomys fimbriatus* were found using the same 27 tree species; however, *P. petaurista* frequented them more often due to a higher level of folivory [117]. Understanding whether population dynamics are impacted by this co-existence is important, as for example when slow lorises and tarsiers live sympatrically, tarsier abundance is lowered [26]. There were 10 more observations of red giant flying squirrels and these are commonly spotted by tourists on night-walks. However, they utilise the nest boxes along the canopy walkway and thus chances of detection are increased. Therefore, knowing the sleep sites for *Aeromys tephromelas* would be an important next step for understanding their activity cycles. Further ecological information on feeding, nesting and home ranges would also help our understanding their interspecific interactions and resource competition.

## 5. Conclusions

Knowledge of flying squirrels is severely limited, with the literature skewed towards the three *Glaycomys* spp. present in North America and *Pteromys volans* distributed across the Northern band of Eastern Europe and Asia [1]. To combat this, research on Asiatic species is required to provide adequate classification of extinction risk, to ensure mitigation efforts are sufficient.

Albeit not extensive, this study provides a glimpse into the black flying squirrel’s ecology and supports their sympatry with the red giant flying squirrel and the Thomas’ flying squirrel, which should be explored further to understand their role in the ecosystem. Despite seeing black flying squirrels when the night-walks noted that they rarely did, this study still yielded low observation rates, remaining consistent with most nocturnal survey efforts [69] and therefore, further improving detection and monitoring methods is vital.

Furthermore, the first documentation of an audible vocalisation of *Aeromys tephromelas* is a leap towards unveiling the enigma of the black flying squirrel, one step at a time.

## Figures and Tables

**Figure 1 animals-14-03323-f001:**
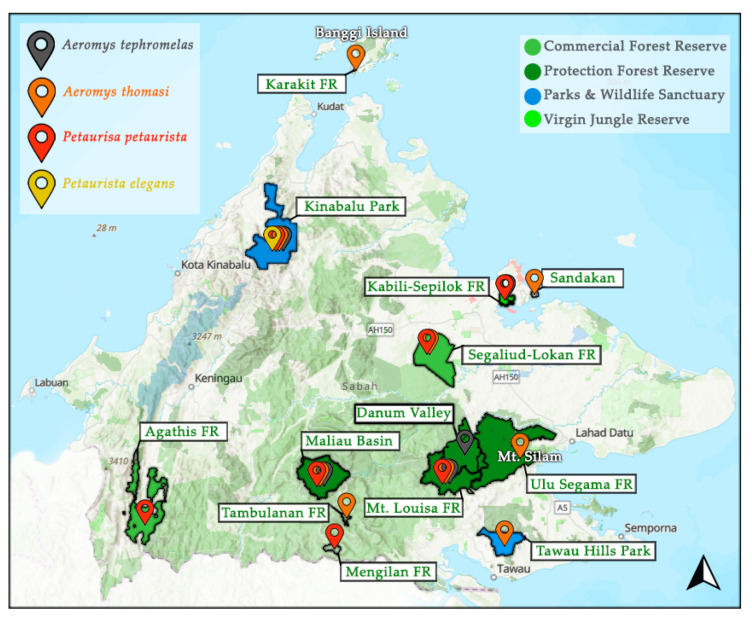
Documented presence of the four giant flying squirrel species in Sabah as mentioned in the literature [6,11,13,15,20,21,22,23,24,25,28,29,30,31,32,33] including: the black flying squirrel (*Aeromys tephromelas*), the Thomas’ flying squirrel (*Aeromys thomasi*), the red giant flying squirrel (*Petaurista petaurista*), and the spotted giant flying squirrel (*P. elegans*). Pin locations are not specific, only general to the area pinned to. FR = Forest Reserve. Map modified using data compiled by Hutanwatch: Sabah Land-use, which provides data on commercial, domestic and protection forest reserves, as well as areas under “parks and wildlife sanctuaries” and virgin jungle reserves.

**Figure 2 animals-14-03323-f002:**
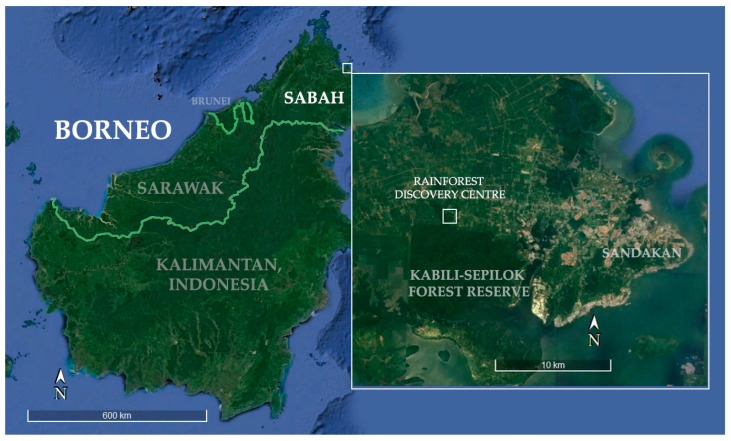
Sabah, Borneo; Zoom in: Kabili Sepilok Forest Reserve with the Rainforest Discovery Centre (5°52′33.84″ N 117°56′38.22″ E) highlighted (Modified from Google Earth Pro© 2024).

**Figure 3 animals-14-03323-f003:**
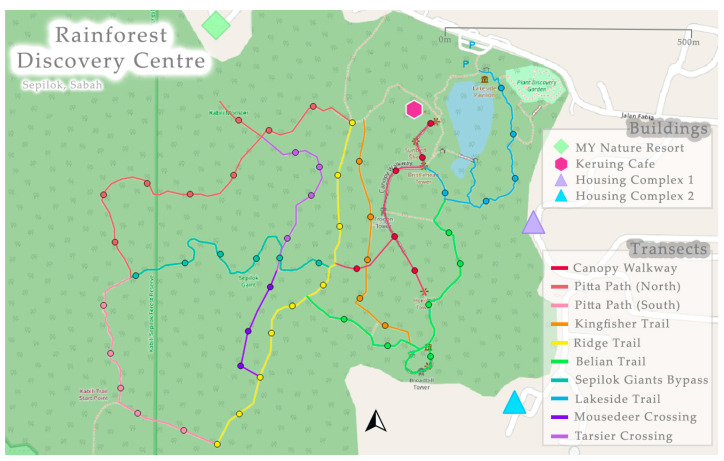
Transects used at the Rainforest Discovery Centre, Sepilok (5°52′33.84″ N117°56′38.22″ E). Transect names are the same as the trails, with the exception that the Pitta path was divided into two transects.

**Figure 4 animals-14-03323-f004:**
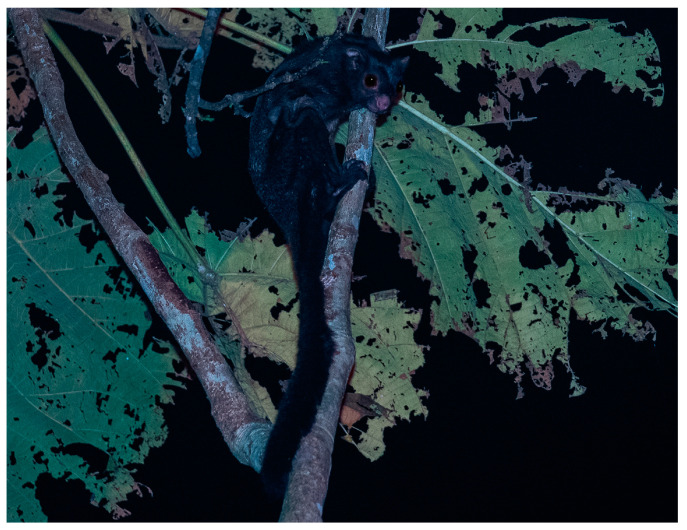
A black flying squirrel observing its surroundings at the Rainforest Discovery Centre. Photograph was taken using a diffuse side flash under red spotlights © Priscillia Miard, March 2023.

**Figure 5 animals-14-03323-f005:**
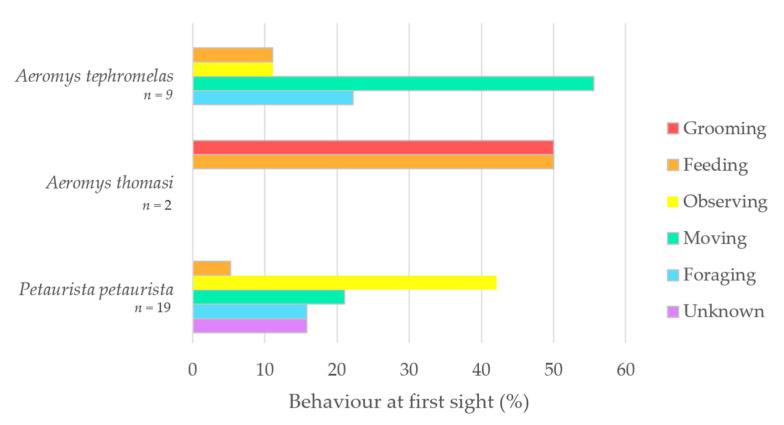
Behaviour at first sight (%) plotted as a bar chart for the black flying squirrel (*Aeromys tephromelas*) (*n* = 9), the Thomas’ flying squirrel (*A. thomasi*) (*n* = 2) and the red giant flying squirrel (*Petaurista petaurista*) (*n* = 19).

**Figure 6 animals-14-03323-f006:**
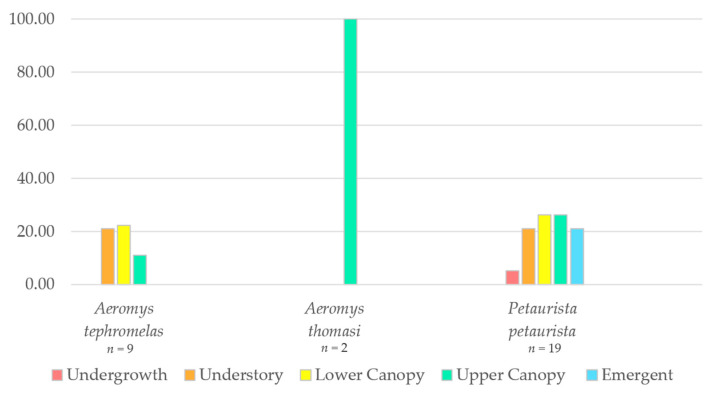
Proportion of observations of the red giant flying squirrel (*Petaurista* petaurista) (*n* = 19), the black flying squirrel (*Aeromys tephromelas*) (*n* = 9) and the Thomas’ flying squirrel *(Aeromys thomasi*) (*n* = 2) within the different rainforest strata.

**Figure 7 animals-14-03323-f007:**
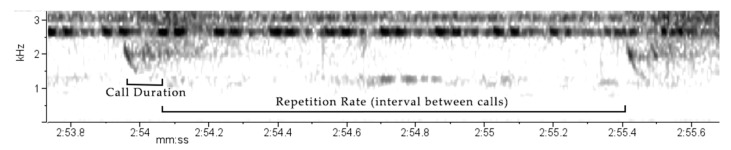
Spectrogram for the call produced by the black flying squirrel (*Aeromys tephromelas*) showing acoustic and temporal variables of three calls. Spectrogram parameters: DFT of 256 samples with Hanning window functioning.

**Table 1 animals-14-03323-t001:** Species of flying squirrel in Sabah, Malaysia including their IUCN status and the most recent assessment date.

Common Name	Species	IUCN Status
Black flying squirrel (giant)	*Aeromys tephromelas*	DD (2016) [6]
Horsfield’s flying squirrel	*Iomys horsfeldii*	LC (2016) [7]
Hose’s flying squirrel	*Petaurillus hosei*	DD (2016) [8]
Jentink’s flying squirrel	*Hylopetes platyurus*	DD (2017) [9]
Red-cheeked flying squirrel	*Hylopetes spadiceus*	LC (2016) [10]
Red giant flying squirrel	*Petaurista petaurista*	LC (2016) [11]
Smoky flying squirrel	*Pteromyscus pulverulentus*	EN (2016) [12]
Spotted giant flying squirrel	*Petaurista elegans*	LC (2016) [13]
Temmink’s flying squirrel	*Petinomys setosus*	VU (2016) [14]
Thomas’ flying squirrel (giant)	*Aeromys thomasi*	LC (2016) [15]
Whiskered flying squirrel	*Petinomys genibarbis*	VU (2016) [16]

**Table 2 animals-14-03323-t002:** Manufacturer and model of the equipment used, alongside their purpose in this study.

Manufacturer	Model	Purpose
Pulsar (Roubaix, France)	Helion XQ 38F	Thermal imaging monoscope
Wolfeyes (Sydney, Australia)	Dingo (800 Lumen)	Red spotlight (headtorch)
Panasonic (Bracknell, UK)	Lumix DMC-TZ61	Digital camera (video recording)
Petterson Elektronik AB (Uppsala, Sweden)	M500-384 USB	Ultrasonic recorder
MiLESEEY (Shenzhen, China)	PF260 Rangefinder	Rangefinder
Garmin (Southhampton, UK)	GPSMAP 64S	GPS
BT Meter (Zhuhai, China)	100 WM	Barometer, Anemometer, Thermo-hygrometer

**Table 3 animals-14-03323-t003:** Flying squirrel ethogram adapted from Ando et al. (1984) [75] and Muul (1965) [76].

Behaviour	Description
Resting	Stationary in a sitting position not interacting with their surroundings, can have eyes closed
Observing	Stationary in a sitting or standing position with active observation of their surroundings
Grooming	Licking and cleaning of own fur (autogrooming)
Feeding	Consumption of food
Foraging	Movement with active search of food, either visual or olfactory
Moving	Continuous and locomotion by running or jumping from one location to another with no discernible purpose
Gliding	Locomotion in air using the patagium to glide from one substrate to another
Fleeing	Interruption of natural behaviour resulting in instantaneous and rapid movement away from the source of the threat
Vocalisation	Vocalisation can be heard from the individual
Unknown	Behaviour cannot be discerned

**Table 4 animals-14-03323-t004:** Observation data of the large black flying squirrel (*Aeromys tephromelas*), the red giant flying squirrel (*Petaurista petaurista*) and the Thomas’ flying squirrel (*A. thomasi*) including the detection method (eyeshine from red spotlight, thermal imaging or from the microphone). The radial distance (m) to observer is included as well as the height in the tree (m) the animal was first observed at.

Species	Date	Time	Point or Opp	Transect	First Detection (Eyeshine, Thermal or Acoustic)	Radial Distance to Observer (m)	Height in Tree (m) (Tree Height m)
*Aeromys tephromelas*	7 February 2023	22:18	Point	RT	Eyeshine	4	8 (10)
*Aeromys tephromelas*	8 February 2023	22:56	Opp	KF	Eyeshine	10	9 (10)
*Aeromys tephromelas*	8 February 2023	23:26	Point	KF	Thermal	3	15 (20)
*Aeromys tephromelas*	9 February 2023	20:56	Opp	LT	Eyeshine	11	11 (12)
*Aeromys tephromelas*	10 February 2023	23:04	Point	RT	Eyeshine	16	10 (40)
*Aeromys tephromelas*	10 February 2023	20:40	Point	RT	Eyeshine	1	25 (40)
*Aeromys tephromelas*	15 February 2023	22:09	Point	RT	Eyeshine	12	8 (10)
*Aeromys tephromelas*	15 February 2023	22:03	Opp	RT	Eyeshine	8	8 (10)
*Aeromys tephromelas*	7 March 2023	20:00	Point	TC	Eyeshine	5	20 (30)
*Petaurista petaurista*	3 February 2023	20:25	Point	CW	Eyeshine	1	8 (10)
*Petaurista petaurista*	3 February 2023	21:32	Point	CW	Eyeshine	4	12 (15)
*Petaurista petaurista*	3 February 2023	20:31	Point	CW	Eyeshine	8	16 (18)
*Petaurista petaurista*	6 February 2023	21:46	Opp	BT	Eyeshine	5	16 (25)
*Petaurista petaurista*	6 February 2023	21:45	Opp	BT	Eyeshine	18	23 (24)
*Petaurista petaurista*	7 February 2023	23:00	Point	RT	Eyeshine	18	20 (22)
*Petaurista petaurista*	8 February 2023	21:10	Point	CW	Eyeshine	4	35 (50)
*Petaurista petaurista*	9 February 2023	20:55	Opp	LT	Eyeshine	11	11 (30)
*Petaurista petaurista*	9 February 2023	22:24	Point	BT	Eyeshine	12	25 (35)
*Petaurista petaurista*	13 February 2023	20:49	Point	CW	Eyeshine	5	8 (9)
*Petaurista petaurista*	8 February 2023	21:31	Opp	CW	Eyeshine	8	24 (24)
*Petaurista petaurista*	1 March 2023	21:36	Point	MD	Eyeshine	2	20 (20)
*Petaurista petaurista*	1 March 2023	21:14	Point	MD	Thermal	5	25 (25)
*Petaurista petaurista*	1 March 2023	21:35	Point	MD	Eyeshine	38	35 (40)
*Petaurista petaurista*	1 March 2023	21:19	Point	MD	Eyeshine	34	40 (55)
*Petaurista petaurista*	1 March 2023	21:18	Point	MD	Eyeshine	38	45 (50)
*Petaurista petaurista*	7 March 2023	20:41	Point	TC	Eyeshine	1	19 (20)
*Petaurista petaurista*	8 March 2023	21:02	Point	PPS	Eyeshine	5	3 (25)
*Petaurista petaurista*	14 March 2023	21:38	Point	TC	Eyeshine	19	25 (50)
*Aeromys thomasi*	3 February 2023	20:46	Point	CW	Eyeshine	2	27 (30)
*Aeromys thomasi*	8 February 2023	21:11	Point	CW	Thermal	6	27 (30)

**Table 5 animals-14-03323-t005:** Sources of food observed during feeding occasions across the survey period for the three giant flying squirrel species. The common name, species, tree family, tree species and the part of the plant fed on are included.

English Name	Species	Tree Family	Tree Species	Part Fed On
Black flying squirrel	*Aeromys tephromelas*	*Meliaceae*	*Heynea trijuga*	Unknown
		*Meliaceae*	*Azadirachta excelsa*	Buds
Thomas’ flying squirrel	*Aeromys thomasi*	*Dipterocapaceae*	*Parashorea tomentella*	Buds
Red giant flying squirrel	*Petaurista petauristsa*	*Rubiaceae*	*Neolamarckia cadamba*	Unknown
		*Sapotaceae*	*Palaquium dasphyllum*	Leaves

**Table 6 animals-14-03323-t006:** Call parameters of 60 vocalisations from one black flying squirrel (*Aeromys tephromelas*). Parameters include the mean and range of call duration (seconds), lower frequency (kHz), upper frequency (kHz), frequency range per individual call (kHz), peak frequency (kHz) and bandwidth at 90% (kHz).

Call Parameters	Mean	Range
Call Duration (s)	0.14	0.08–0.18
Lower Frequency (kHz)	1.19	0.75–1.62
Upper Frequency (kHz)	2.37	2.11–2.69
Frequency Range (kHz) (per individual call)	1.19	0.78–1.72
Peak Frequency (kHz)	1.97	1.89–2.58
Bandwidth (90%) (kHz)	0.69	0.48–1.98

## Data Availability

The original contributions presented in the study are included in the article and Appendix A; further inquiries can be directed to the corresponding author.

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
