# Peer review of "Effective Survey Methods for the Elusive Data Deficient Black Flying Squirrel (Aeromys tephromelas) in Sabah, Malaysia Facilitate First Vocalisation Record"

_animals, 2024, doi:10.3390/ani14223323_

Round 1
Reviewer 1 Report
Comments and Suggestions for Authors
The manuscript “First vocalisation record of the elusive ‘Data Deficient’ black flying squirrel (Aeromys tephromelas) in Sabah, Malaysia” describes a study conducted in the Rainforest Discovery Center in Sepilok, Sabah. The study involved surveys for flying squirrel species conducted across a 2-month period using several different survey methods.
Given the paucity of published research on the black flying squirrels, and flying squirrel species in general in this geographic area, these findings will be of interest to those studying nocturnal species behavior and ecology (the focus of this special issue). Below I have outlined some general comments and then some specific comments by line number.
General comments:
Title: I might consider adding something about survey methods to the title. Although the recording of the vocalization is very interesting, the paper focuses on the use of red spotlight to survey the squirrels along the points and transects. Perhaps something like “Effective survey methods for the elusive ‘Data Deficient’ black flying squirrel (Aeromys tephromelas) in Sabah, Malaysia facilitates first vocalisation record”
Methods: It would be nice to have a bit more detail in several areas to increase clarity of what data were collected and how. I would suggest having a Data Collection subsection where you include a description of all the data collected (behavioral, vocalizations, and covariates) and then in the Data Analysis section focus on how the data were summarized and analyzed.
Common versus Scientific names: Throughout the paper, you switch between using the common and scientific names. I am not sure what the journal editors prefer, but it would likely be best to use one format and then be consistent.
Specific comments:
Abstract Lines 33: Change to “sheds light”
Lines 49-52: At first I thought perhaps the number of papers published was something the authors generated, but then I saw the citation and realized those numbers came from the Koprowski and Nandini 2008 paper. Given that was ~14 years ago (the literature review they did was to Jan 15 2006), I think it might be useful to say that a literature review through 2006 found those numbers. More studies have been published since 2006.
Line 63/97: Change classed to “classified” or “categorized”
Lines 158-160: Might also want to have a look at this paper: Newar, S. L., Schneiderová, I., Hughes, B., & Bowman, J. (2024). Ultrasound and ultraviolet: Crypsis in gliding mammals. PeerJ, 12, e17048.
Lines 158-163: What was the Pettersson Elektronik M500-384 USB microphone attached to for the surveys, and could you also provide the settings used?
Table 2: Change “Manufacturer” to Behaviour
Lines 194-197: It is not clear what is meant by an “ad hoc observation”. Also, it seems like this would go in the results?
Figure 3. I would move this to be after the paragraph where the figure is referred to
Lines 223-225: Could you summarize how many of the 19 and 2 observations of the other species were with the eyeshine or thermal method?
Lines 227-228: Need to include Aeromys thomasi in this statement
Figure 4: Can you include a “n = #” for each of the species either in the graph or in the figure caption
Line 256: Also include year with date
Lines 259-260: Why was only 3 min and 20 sec able to be analyzed?
Line 263: Add comma after minutes
Lines 286-287: Rephrase this statement for clarity
Lines 294-296: This statement is not clear as only 3 FR are listed not 4. It might also be easier to visualize if you say the number of forest reserves inventoried rather than 10% of 333 (which is 33.3, so unclear if it is out of 33 or 34).
Lines 294-298: Since you have not given detection rates for the other forest reserves, it seems like this statement (specifically that detection was higher in this study) is not the correct comparison. Can you provide a form of detection rate for the 4 forest reserves where they were detected?
Lines 303-304: Need to clarify if this statement “For flying squirrels, the only activity budgets published focus on the Indian giant flying squirrel (Petaurista philippensis), which has two nightly peaks between 19:00-20:00 and 03:00-04:00 [59,60].” is referring to species in Asia (or a smaller geographic range). Or, if referring to all flying squirrel species, include references for activity budget work done on North American and Eurasian species.
Lines 305-306: to be consistent, perhaps use the 24 hour clock format throughout
Lines 312-313: Change to “The only mention about flying squirrel activity and the influence of moonlight in Asia is for Eoglaucomys fimbriatus (Kashmir) where they hypothesised that moonlight did not impact their activity [63].”
Line 340: Since P. Miard is an author, perhaps switch this to Personal Observation?
Line 397: I assume you also meant to cite reference 81 in addition to 82 here
Lines 420-421: This statement is not clear – a lack of vocal response? Also, the 42% of flying squirrel species with known calls not having an associated behavior while vocalizing is likely more due to not having observations of the context in which the call are given, rather than the vocalizations not producing an associated behavior
Line 427: Might make clearer if you move the reference to be after “described calls”
Reviewer 2 Report
Comments and Suggestions for Authors
I am fine with the simple summary but I think the title has to be changed. It does not describe the material presented in this paper adequately.
The abstract covers quite different subjects as described in the simple summary. It should contain all aspects covered in the paper.
Introduction
I am fine with the general part about flying squirrels, the description of the species and the red list status. I do not think that the distribution map is necessary and also the description of the Sabah situation especially because the reader does not know yet, what Sabah is. This part could be shifted to the study site. If you keep Fig. 1 please provide the reader with the information about the spatial situation of Sabah. I am also fine with the description of the problems surveying flying squirrels.
Line 109-115 This part contains some results and should be formulated as research question. Now it is more a summary of what you did.
I am fine with the description of the study site but would like to see a map about the position of the site.
I am fine with the description of the surveys. Fig. 2 – I think to location of the transects is information enough. In my opinion no names are need and if you like to provide the reader with the position of the point counts it is not necessary to name them. Lin e147-149 is not necessary to understand the figure.
Table 1 – you can include this information in the text.
Line 202-205 These results were not presented
Line 205-207 is part of the results. Do not mention it in the paper or describe the method and mention it in the results.
Results
Is it possible to include the data for the other two species in table 3?
In my opinion it does not make sense to plot the behaviour of the different species (Fig. 4) because of low sample size. Information is included in Table 3.
It is the same for Fig 5 but maybe it makes more sense here. Line 246 - (understorey, undergrowth, lower canopy, upper canopy, emergent). Is not necessary. This is already included in the graph. Does it make sense to test the different distributions?
Table 4 – I would rethink the layout. It is not clear what tree species belongs to what squirrel species.
I am fine with bioacoustics
Discussion
I am fine with the first part of the discussion.
Research recommendations are based on very low sample size and focuses mainly on RDC. Change in methods are already discussed and recommendations for a specific site should not be part of a scientific paper.
The conclusions should conclude the results and not promote a site.
I think it is important to publish data and observations of species which are nearly unknow and I understand that this is a first step to gain more knowledge. As a result, the sample size is very low, the paper is very descriptive and touches a lot of aspects. This is okay for a paper with a focus on faunistics.
Round 2
Reviewer 2 Report
Comments and Suggestions for Authors
Simple summary – even with the summary you should stick with the structure introduction, method, results, discussion.
In my opinion the new title does not describe the material presented in this paper adequately. I suggest - Effective survey methods for giant flying squirrels, especially the elusive ‘Data Deficient’ black flying squirrel (Aeromys tephromelas) in Sabah, Malaysia
I am fine with the abstract
Introduction
1) Overlook flying squirrels in general
2) Flying squirrels in Sabah, description of these species
3) Ecological role of flying squirrels
4) Problems with survey
5) Acoustic communication – acoustic monitoring
In my opinion the introduction needs restructuring. Put the information in the section where it has to be. If you describe the topics above in general, I am fine with it and the reader gets all the information needed. I see no need to cover Sabah as a topic in the introduction. The Sabah situation has to bee shifted to the study area, even the distribution map and some topics should be considered in the discussion.
I disagree with the statement “As it is a descriptive paper, there is not a research question and often in these cases a summary is used rather than a question/aim”.
There is a question – what species are present in Sabah, what are they feeding on and how do the use their habitat. Line 213-216 is a result and should be shifted to this section.
I am fine with the description of the study site.
I am fine with the description of the surveys. Fig. 3 – I think to location of the transects is information enough. In my opinion no names are need and if you like to provide the reader with the position of the point counts it is not necessary to name them. I disagree with the statement: “
We believe the point count names are useful as a guide for the reader as when describing the location of the vocalisation, the figure can be referred to.” To make it more clear in-text we have added (Figure 3) after “the Ridge trail” (Line 152) – I could not find this section.
Results – I am fine with this section
Discussion, conclusion - I am fine with these sections
